# Radiological correlates of vocal fold bowing as markers of Parkinson's disease progression: A cross-sectional study utilizing dynamic laryngeal CT

**Andrew Ma**[1,2,3], **Kenneth K. Lau**[4,5], **Dominic Thyagarajan**[1,2,3]*

**1** Department of Neurosciences, Central Clinical School, Monash University, Melbourne, Australia, **2** Department of Neurology, Monash Health, Melbourne, Australia, **3** Department of Neurology, Alfred Health, Melbourne, Australia, **4** School of Clinical Sciences, Faculty of Medicine, Nursing and Health Sciences, Monash University, Melbourne, Australia, **5** Monash Health Imaging, Monash Health, Melbourne, Australia

* dominic.thyagarajan@monash.edu

## Abstract

**Data Availability Statement:** Data will be publically available in the figshare public repository at http://

### Objective

To determine whether arytenoid cartilage position and dynamics change with advancing duration and severity (as graded by MDS-UPDRS part III scores) in Parkinson's disease, in a cross-sectional study design, we performed laryngeal four-dimensional computed tomography (4D-CT) in people with Parkinson's disease and controls.

### Methods

31 people with Parkinson's disease covering a range of disease duration and severity and 19 controls underwent laryngeal 4D-CT whilst repeatedly vocalizing. We measured on each CT volume the glottic area (*GA*), inter-arytenoid distance (*IAD*), IAD-Area index (*IAI*) and arytenoid cartilage velocity ($\overrightarrow{av}$).

### Results

People with Parkinson's disease had reductions in the mean/effective minimum *IAD* when compared to controls, while mean/effective minimum *GA* and mean/effective maximum *IAI* were increased. Arytenoid cartilage velocities showed no difference. On Spearman correlation analyses, advancing disease duration and severity of PD showed moderately strong and significant correlations with increasing mean/effective minimum *GA*, increasing mean/effective maximum *IAI* and decreasing effective minimum *IAD*. Linear mixed models which considered the effects of intra and inter-individual variation showed that both disease duration (b = -0.011, SEb = 0.053, 95% CI [-0.022, 0], t(27) = -2.10, p = 0.045) and severity (b = -0.069, SEb = 0.032, 95% CI [-0.14,-0.0039], t(27) = -2.17, p = 0.039) were significant predictors for *IAD*, and also for transformed values of the *GA* and *IAI*.

doi.org/10.26180/14214056 (available after acceptance of manuscript).

**Funding:** A. Ma received salary from Ipsen Biopharmaceuticals as support for a Movement Disorders Fellowship and received support from an Australian Government Research Training Program (RTP) Scholarship. The funders provided support in the form of salaries for author AM, but did not have any additional role in the study design, data collection and analysis, decision to publish, or preparation of the manuscript. The specific roles of these authors are articulated in the 'author contributions' section.

**Competing interests:** A. Ma received salary from Ipsen Biopharmaceuticals as support for a Movement Disorders Fellowship and received support from an Australian Government Research Training Program (RTP) Scholarship. The funders had no role in study design, data collection and analysis, decision to publish, or preparation of the manuscript. D. Thyagarajan and K.K. Lau report no disclosures relevant to the manuscript. This does not alter our adherence to PLOS ONE policies on sharing data and materials."

## Conclusions

There are progressive alterations in phonatory posturing as Parkinson's disease advances. The increases in *GA* despite reductions in *IAD* are concordant with prior observations of vocal fold bowing. Our study provides a basis for using laryngeal 4D-CT to assess disease progression in Parkinson's disease.

## Introduction

Parkinson's disease (PD) is a neurodegenerative condition which remains a significant health issue worldwide. No current intervention alters the course of the disease. Lack of biomarkers of disease progression hampers the search for disease-modifying treatments. For lack of alternatives, assessing progression in PD is still primarily focused on clinical assessment of motor function, often with the aid of structured rating scales such as the Unified Parkinson's Disease Rating Scale (UPDRS). Yet, these assessments are subjective with high inter- and intra-rater variability [1, 2]. More reliable assessment tools would allow for shorter follow-up periods and smaller sample sizes when studying the effects of novel disease-modifying therapies [3].

Voice disorder affects 70–90% of people with Parkinson's disease (pwPD) [4–6]. There is evidence that voice dysfunction is the earliest sign of motor impairment in PD [7–9]. Despite this, perceptual, acoustic or other measures of hypokinetic dysarthrophonia in PD (e.g. vocal cord dysfunction) have not yet found a place in the assessment of disease progression.

Laryngeal computed tomography (CT) is a potential technique to study these voice changes. Our previous study [10] used laryngeal CT during vocalization to measure glottic area (*GA*) and arytenoid cartilage position. It found differences in the inter-arytenoid distance (*IAD*) between patients with early PD and healthy controls. In that study, participants also underwent perceptual analysis and we found significant increases in breathiness and articulatory dysdia-dochokinesis, and reductions in loudness variability and mean phonation time in Parkinson's disease. We found no effect of age or sex on *IAD* or *GA*. However, that study provided no insight into how these measures change with disease duration and severity. Yet, such objective measures of vocal cord dysfunction would be valuable in measuring the effects of potential disease-modifying therapies.

Based on our prior findings, our hypotheses are that disease duration or motor severity is: 1) positively correlated with mean and effective minimum *GA*, 2) negatively correlated with mean and effective minimum *IAD*, and effective maximum arytenoid cartilage velocities. The aim of this study is to characterise how these laryngeal measures (*GA* and *IAD*) change as PD progresses.

## Materials and methods

### Patients and recruitment

We previously studied 19 healthy controls and 15 pwPD. Those patients had a disease duration less than 6 years and modified Hoehn and Yahr (H&Y) stage of 2.5 or lower [10]. We expanded this cohort by recruiting patients with more advanced PD from the Movement Disorders Clinic at Monash Medical Centre. For the purposes of obtaining adequate representation across the spectrum of the disease, we recruited patients who were H&Y stage 3 or more and a disease duration of 5 years or greater. Diagnoses of idiopathic PD were made in accordance with the UK Brain Bank Criteria. Control participants either responded to

advertisements or were spouses of the pwPD who were unaffected by Parkinson's disease or other neurological disorders apparent on clinical assessment by a neurologist.

## Standard protocol approvals, registrations and patient consents

In a cross-sectional design, all participants were imaged only once. Written informed consent was obtained from all participants according to the Declaration of Helsinki. Ethics approval was granted by the Research Ethics Committee of Monash Health (HREC reference number 11230B).

## Image acquisition and analysis

Four-dimensional imaging data with an anatomical z-axis of 16 cm over the larynx was acquired using a 320 multi-detector row CT (Aquilion One, Tokyo, Canon Medical Systems). Participants were imaged in the supine position without CT table movement. During a continuous CT acquisition scanning period of 5 seconds, participants were instructed to produce five short phonations of /i/ quickly and clearly at a comfortable speaking volume and pitch. Patients practiced the vocalization task prior to undergoing the scan. Multiple phonations were performed to allow study of the arytenoid cartilages in motion in as repeatable a way as possible. Imaging acquisition was terminated before five seconds in those who had completed the vocalization task early. Images were re-constructed to produce continuous multiplanar images of the larynx at 100ms per frame. pwPD were assessed in the practically-defined 'off' state by withholding their regular PD medications overnight. In the newly recruited participants, fiducial coordinates of the arytenoid cartilages were obtained using the open-source ImageJ software [11].

## Statistical analysis

Data transformation, statistical analysis and graphics were performed using the R statistical and graphing package [12] and nlme [13].

In pre-processing, we manually removed *IAD* and *GA* values from the pre and post-vocalization period. Whilst not vocalizing, values of both the *GA* and *IAD* are strikingly higher as the vocal folds are apart. Trimming was performed by visually inspecting the raw *GA* and *IAD* data sets and excluding periods containing these larger values. This was done as the study was focused on assessing the movement and posturing of the vocal folds during vocalization.

The raw data set consisted of *GA* and fiducial markers on each arytenoid cartilage for each timepoint, with successive timepoints at 100 msec intervals. Workstation software (IntelliSpace Portal, Philips Healthcare, Cleveland, USA) was utilised to adjust the continuous dynamic CT images to the plane of the vocal folds. The glottic area (GA) was then segmented at each 100ms frame. *IAD* was the calculated Euclidean distance between the fiducial markers on the left and right arytenoid cartilages. These measures are illustrated in Fig 1. We also calculated a dimensionless index, the IAD-Area index (*IAI*), which we defined as $\frac{\sqrt{GA}}{IAD}$. The instantaneous velocity of arytenoid cartilage movements ($\overrightarrow{av}$) was calculated as the sum of the distance in mm that the left and right fiducial markers moved between successive 100ms timepoints. We also separated these values as either representing abduction velocity ($\overrightarrow{abv}$) or adduction velocity ($\overrightarrow{adv}$), based on the direction of arytenoid cartilage movement over that time interval.

We defined an 'effective minimum' for *IAD* and *GA* as the median of the lowest five values, as the vocalization task involved phonating /i/ five times. Similarly, the 'effective maximum' *IAI, AS*, $\overrightarrow{abv}$ and $\overrightarrow{adv}$ was defined as the median of the highest five values. This was done to minimise the potential distorting effects of outliers.

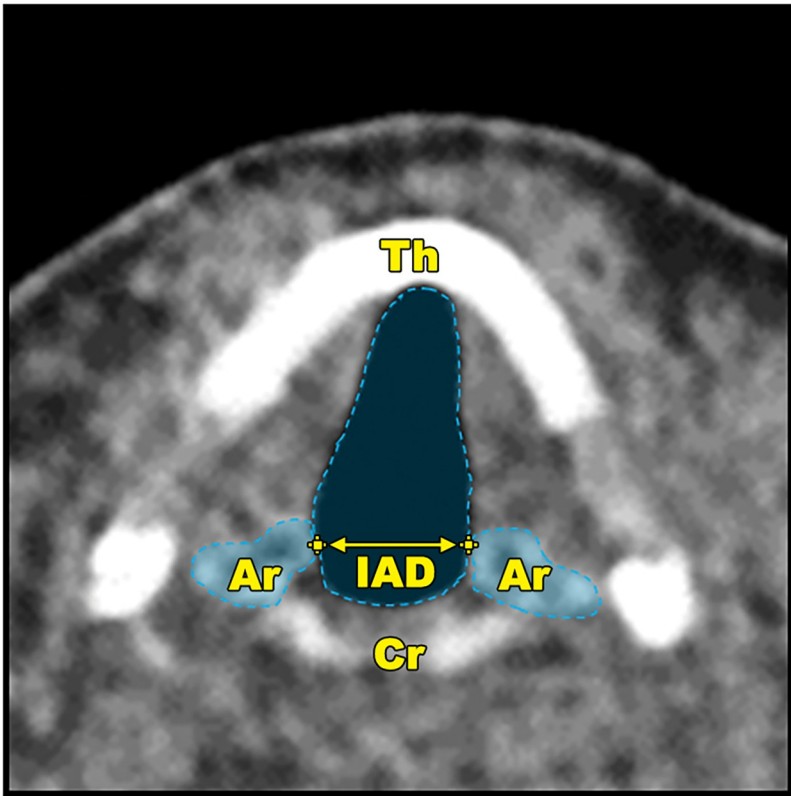

**Fig 1. The laryngeal measures.** Annotated CT image at the level of the glottis depicting the laryngeal measures used in this study. Fiducial markers (shown as yellow crosshairs) are placed at the most medial aspect of the arytenoid cartilages (Ar). The inter-arytenoid distance (*IAD*) is defined as the Euclidean distance between them, as depicted by the arrow. The arytenoid velocity ($\overrightarrow{av}$) is equivalent to the change in *IAD* between successive timepoints (not illustrated). The glottic area (*GA*) is the segmented area of air density at the level of the vocal folds. Also shown are the thyroid cartilage (Th) and cricoid cartilage (Cr).

Statistical tests utilised the mean *GA* and *IAD* (analogous to our previously reported key measures $AUC_{ga}$ and $AUC_{iai}$ respectively) [10], effective minimum *GA* and *IAD*, mean IAI and $\overrightarrow{av}$, and the effective maximum *IAI*, $\overrightarrow{av}$, $\overrightarrow{abv}$ and $\overrightarrow{adv}$. We set an alpha level of 0.05 for all statistical tests. Wilcoxon's rank sum test was then used to compare the above against condition (PD/control).

**Table 1. Demographic details.**

|  | Controls | PD |
| --- | --- | --- |
| n | 19 | 30 |
| F:M | 8:11 | 9:20 |
| Mean age in years | 70.8 | 69.7 |
| SD age in years | 7.38 | 7.32 |
| Median disease duration (months) | NA | 83.5 |
| Lower bound IQR of disease duration (months) | NA | 48.8 |
| Upper bound IQR of disease duration (months) | NA | 138 |
| Median UPDRS Part III | 0.0 | 19.5 |
| Lower bound IQR of UPDRS Part III | NA | 15 |
| Upper bound IQR of UPDRS Part III | NA | 30.8 |

Spearman's rank correlation analysis as well as linear mixed effects modelling were performed to determine how these measures relate to PD duration (in months) and severity (as graded by UPDRS part-III scores). Models considered the laryngeal measures as the dependent variable, with non-linear transformations applied to the *GA* and *IAI* to achieve linearity of the data. Disease duration, UPDRS part-III scores, age and sex were considered as fixed effects. The intercepts were considered as a random effect to account for inter-participant variability.

## Results

### Study population and baseline characteristics

Since the prior study, we recruited 14 further pwPD, with the overall cohort consisting of 31 pwPD and 19 controls. Their baseline characteristics are presented in Table 1. No pwPD recruited had previously undergone voice training. One patient was excluded because gross imaging artefact precluded accurate image analysis. Another patient was excluded from analysis of arytenoid motion as the images were inadvertently captured at 300 ms/frame, rather than our standard of 100 ms/frame. After exclusion of these patients and data trimming, the total number of timepoints analysed was 1671 for the *IAD*, 1705 for the *GA* and 1607 for the *IAI*.

### Arytenoid cartilage position and glottic area is altered in Parkinson's disease

In comparing participants with PD and controls, statistically significant differences were found in the median values across the vocalization period of the mean *IAD* (*Mdn* = 4.24 vs 5.21, *p* = 0.006), mean *GA* (*Mdn* = 53.0 vs 21.2, *p* = 0.024) and mean *IAI* (*Mdn* = 1.54 vs 0.852, *p* = 0.001). Significant differences were also seen for the effective minimum *IAD* (*Mdn* = 2.66 vs 3.69, *p* = 0.009), effective minimum *GA* (*Mdn* = 23.2 vs 3.60, *p* = 0.002) and effective maximum *IAI* (*Mdn* = 2.68 vs 1.35, *p* < .001). Mean and effective maximum $\overrightarrow{av}$, as well as effective maximum $\overrightarrow{abv}$ and $\overrightarrow{adv}$, all were not significantly different in pwPD when compared to controls (see Fig 2 and S1 Table).

### The IAD, GA and IAI is correlated with the duration and severity of Parkinson's disease

Plots of the mean/effective minimum *GA* and mean/effective maximum *IAI* show that these measures increase with disease duration and UPDRS part-III scores, while mean/effective minimum *IAD* decrease. As there was no significant difference seen in the $\overrightarrow{av}$, $\overrightarrow{abv}$ and $\overrightarrow{adv}$ in those with pwPD when compared to controls, these measures were not included in the correlation analyses.

Spearman correlation analysis demonstrates moderately strong and significant correlations between all laryngeal measures, except for the mean *IAD*, with both disease duration and UPDRS. The strongest correlation was between the effective minimum *GA* and disease duration (Spearman Rho = 0.692) (see Fig 3).

Linear mixed regression models showed statistically significant effects of both the duration and severity of PD on the *IAD*, as well as transformed values of the *GA* and *IAI*. Non-linear transformations were applied to the *GA* and *IAI* to achieve linearity of the data set. This allowed all models to meet all the standard statistical assumptions. Duration as a fixed effect significantly predicted the *IAD* with b = -0.011, SEb = 0.053, 95% CI [-0.022, 0], t(27) = -2.10, p = 0.045. Similarly, the UPDRS as a fixed effect predicted the *IAD* with b = -0.069, SEb = 0.032, 95% CI [-0.14, -0.0039], t(27) = -2.17, p = 0.039. The transformed values of the

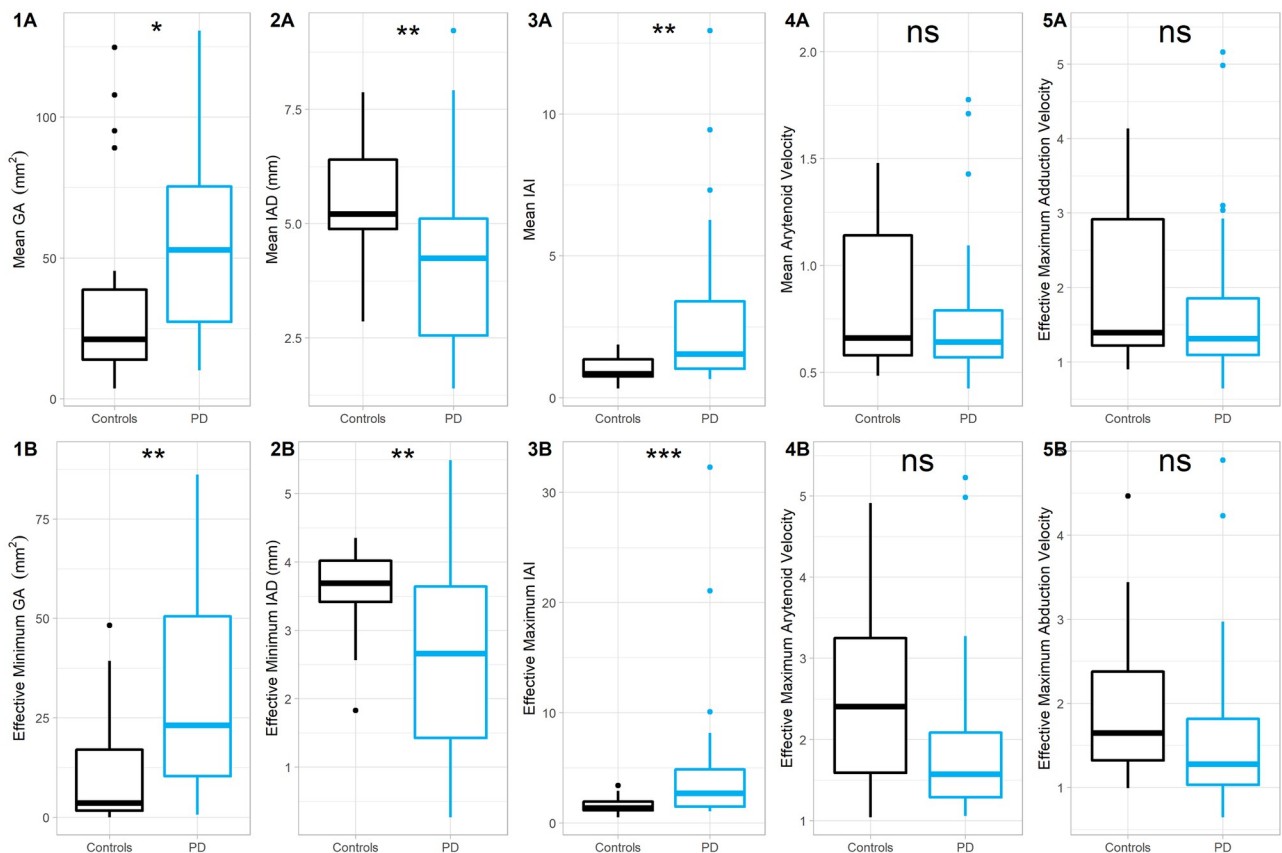

**Fig 2. Comparison of the laryngeal measures during vocalization between pwPD and controls.** Boxplots comparing measures of laryngeal dynamics in participants with Parkinson's disease (blue boxes) against controls (black boxes). (1-3A) Comparison of the means of the *GA*, *IAD* and *IAI* during the period of vocalization. (1-3B) Comparison of the values of the laryngeal measures at maximal vocal fold adduction, corresponding with the effective minimum *GA*, effective minimum *IAD* and effective maximum *IAI*. The mean and effective maximum velocities of the arytenoids is assessed in (4A) and (4B) respectively, calculated by assessing the extent of movement of the arytenoids between successive 100 ms volumes. Sub-analysis of the maximal velocities by the direction of vocal fold movements is shown in (5A) (adduction; $\overrightarrow{adv}$) and (5B) (abduction; $\overrightarrow{abv}$). ns p>0.05; * p<0.05; ** p<0.01; *** p<0.001.

*IAI* in model 3, predicting the duration (b = 0.0054, SEb = 0.0014, 95% CI [0.0024, 0.0083], t(27) = 3.69, p < .001) and model 6, predicting severity (b = 0.031, SEb = 0.0089, 95% CI [0.012, 0.049], t(27) = 3.41, p = 0.002) were the preferred models for interpretation based on having the highest marginal R-squared values (see S2 Table for model details and comparisons). Models were all adjusted to account for the age and sex of participants, although neither demonstrated a significant interaction with the laryngeal measures. Additionally, no significant interaction was found between duration of disease or severity and participant age or sex.

## Discussion

The main findings of this study are that in PD, disease duration and severity are correlated positively with mean/effective minimum *GA* and negatively with the mean/effective minimum *IAD*. This relationship was confirmed after controlling for age and sex with a linear mixed random effects model. Thus, these laryngeal measures have potential utility as a marker of disease progression.

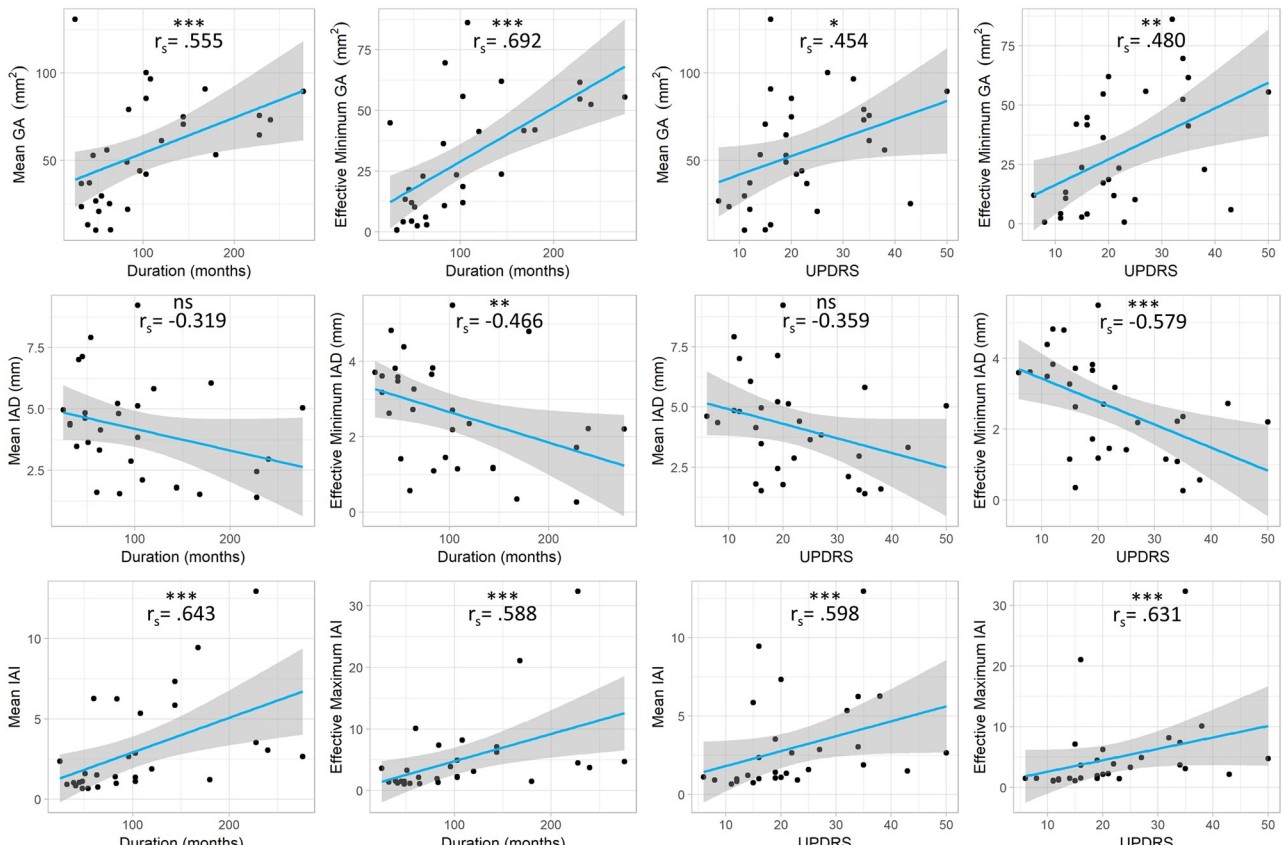

**Fig 3. Change in laryngeal measures with disease duration and UPDRS part III scores.** Plots of the raw data of the laryngeal measures against duration in months and UPDRS part-III scores. The linear regression 'line of best fit' is marked in blue, with the 95% confidence interval shaded in grey. Spearman correlation coefficients ($r_s$) between the laryngeal measures and disease duration or UPDRS are annotated for each. ns $p>0.05$; * $p<0.05$; ** $p<0.01$; *** $p<0.001$.

How do these laryngeal measures relate to previous knowledge of phonatory dysfunction in PD? Prior laryngoscopic studies have identified abnormalities of phonatory posture in pwPD characterized by glottic incompetence due to bowing of the vocal folds, as well as alterations in the position of the arytenoid cartilages, vocal folds and ventricular folds (false cords) [14–16]. One of these studies followed-up patients for up to 4 years but did not identify any change in these laryngeal characteristics on repeat laryngoscopic assessment for any given patient [14]. Our study, possibly owing to the capacity to perform precise anatomical measurements rather than observation alone, has identified change with disease progression.

The increase in mean and effective minimum *GA* in pwPD compared with controls is in keeping with the presence of glottic incompetence in pwPD. This is thought to contribute to the breathiness of the voice that is often clinically appreciable in pwPD [17, 18]. It should be noted that our previous study did not find significant differences in the mean *GA* between PD and healthy controls. However, this study only assessed patients earlier in the course of the disease. As *GA* is correlated with duration of disease, in this cohort including patients with more advanced PD, we found that the mean *GA* is also significantly increased.

With regards to the *IAD*, the mean and effective minimum values are reduced during vocalization in pwPD when compared to controls i.e. the arytenoid cartilages are closer together during vocalization in PD. This is in keeping with laryngoscopic observations that in pwPD

whose motor symptoms are asymmetric, on the side of the body with more severe parkinson-ism, the arytenoid cartilage is hyperadducted, its vocal fold closes underneath the contralateral vocal fold and there is increased contraction of the adductor muscles of the vocal folds. This results in the arytenoid cartilage on the more affected side being positioned with its vocal pro-cess and apex more posterior, and its apex more lateral, when compared with the other side [14]. Their observations are supported by laryngeal EMG studies of pwPD. These studies have shown increased spontaneous muscle activity in the laryngeal adductors (thyroarytenoid, cri-cothyroid and lateral cricoarytenoid muscles) at rest [15, 19–21]. Also, EMG activity of the only abductor of the vocal folds (the posterior cricoarytenoid muscle) is decreased [15]. These laryngoscopic and EMG findings (see Fig 4) explain the hyperadduction of the arytenoid carti-lages seen on laryngoscopy, and which we have now also demonstrated on the 4-dimensional CT.

The change in glottic shape caused by vocal fold bowing was indirectly measured by defin-ing the *IAI*, a ratio which relates *GA* to *IAD*. A disproportionately elevated *GA* for a given *IAD* implies bowing of the vocal folds. Although vocal fold bowing can also occur with ageing due to vocal fold atrophy [22], we showed that the mean/effective maximum *IAI* is increased in pwPD when compared to age-matched controls. The mean/effective maximum *IAI* is more

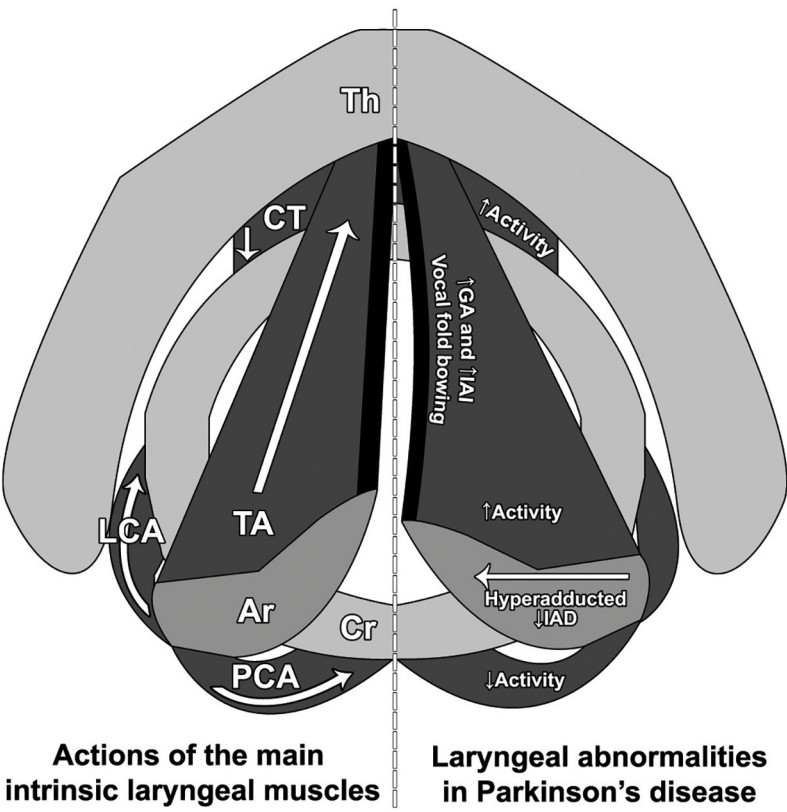

**Fig 4. Actions of the main intrinsic laryngeal muscles and the laryngeal abnormalities in Parkinson's disease.**
Schematic diagrams of the larynx viewed from above. The left panel demonstrates the normal anatomy of the main laryngeal cartilages (Th, thyroid cartilage; Cr, cricoid cartilage; Ar, arytenoid cartilages), as well as the main abductors (LCA, lateral cricoarytenoid muscle; TA, thyroarytenoid muscle) and adductor (PCA, posterior cricoarytenoid muscle), with their primary movements depicted by the accompanying arrows. The right panel highlights the prior laryngoscopic findings, as well as the EMG abnormalities described which account for these. The vocal fold bowing explains the increase in glottic area (*GA*) and IAD-Area Index (*IAI*) seen on laryngeal CT, while hyperadduction of the arytenoids causes the reduction in inter-arytenoid distance (*IAD*).

strongly correlated with UPDRS part-III scores than either the *GA* or *IAD* alone. The preferred linear mixed models based on effect size were those which treated transformed values of the *IAI* to explain disease duration or severity. As the *IAI* considers both the changes to *GA* and *IAD* which occur in the disease, increased *IAI* may also be more specific for PD, although this would need to be confirmed in future studies.

We did not identify the presence of bradykinesia in the arytenoid cartilage movements during vocalization, with the mean and effective maximum $\overrightarrow{av}$ both not significantly reduced in pwPD when compared to control subjects. We also demonstrated hyperadduction of the arytenoid cartilages, with a reduction in *IAD*, suggesting against the presence of arytenoid cartilage hypokinesis. However, our method of image analysis only measures the sliding adduction movements, but we could not easily measure rotational movements of the arytenoid cartilages. To do so, suitable landmarks (e.g. the vocal and muscular processes) would need to be marked with fiducials but these are not always reliably identifiable on CT. Therefore, we cannot exclude the presence of bradykinesia or hypokinesis in arytenoid cartilage movements in other planes.

Effective minimum *GA* and *IAD* during vocalization, which occur at maximal vocal fold adduction, showed significant differences between PD and controls. They also change as the disease progresses, with the *GA* increasing and the *IAD* decreasing, with advancing disease duration and severity. Measuring the effective minimum *GA* and *IAD* reduces variability introduced by differences in how participants performed the vocalization task. Future studies assessing the laryngeal dynamics in PD could consider measuring the *IAD* and *GA* just at the point of maximal vocal fold adduction during vocalization.

Limitations of our study include the possibility of variance being introduced by differences amongst participants in volume, pitch and cadence. Vocalization during image acquisition could not be recorded due to the loudness of the scanning apparatus. As such, we could not account for these acoustic parameters which may influence arytenoid cartilage dynamics.

The presence of vocal tremor was also not assessed. Vocal tremor occurs in some pwPD, with a frequency depending on technique and definition– 13–68% on perceptual studies and about 15–55% on endoscopic studies—but at a similar prevalence to controls on acoustic analysis [23]. The contribution of vocal tremor is therefore uncertain. The addition of a sustained phonation task may have helped us to understand the contribution of vocal tremor, but this would have exposed the subjects to more radiation.

Another limitation of our study is its cross-sectional design. To establish these radiographic laryngeal measures as markers of disease duration or severity, a prospective study would be ideal. Imaging modalities which do not involve the use of ionising radiation such as ultrasound would be preferable in a prospective study. Ultrasound can reliably detect the arytenoid cartilages and the movements of the vocal folds [24, 25] and we are exploring the use of this modality.

In summary, this study has identified radiographic measures which correlate with the anatomic changes previously observed in laryngoscopic studies of pwPD. We conclude that certain measures of laryngeal dynamics change with increasing duration and severity of PD. Therefore, our results demonstrate the utility of dynamic laryngeal CT as a means of objectively tracking the duration and severity of PD. Validating these results in a prospective cohort with other imaging modalities (e.g. ultrasound) is a useful direction in a future study. Further study in atypical parkinsonian conditions, as well as determining the effect of levodopa administration would also prove to be of great interest. The use of imaging modalities may elucidate the disease trajectory and offer opportunities for objective monitoring of PD.

## Supporting information

**S1 Table. Laryngeal measures during vocalization per participant.** Table of summary statistics for each laryngeal measure by patient. Mean *GA*, *IAD* and *IAI* and their standard deviations (SD) during the vocalization period are listed. The effective minimum *GA* and *IAD* and effective maximum *IAI* listed refers to the median of the five lowest or highest values, whilst the inter-quartile range (IQR) given corresponds to the IQR for these five values.
(DOCX)

**S2 Table. Linear mixed models.** Table listing the details of the linear mixed models used to analyse the effect of duration or severity (as graded by UPDRS part-III scores) of PD on the *IAD* and transformed values of the *GA* and *IAI*.
(PDF)

## Author Contributions

**Conceptualization:** Kenneth K. Lau, Dominic Thyagarajan.

**Data curation:** Andrew Ma, Dominic Thyagarajan.

**Formal analysis:** Andrew Ma, Dominic Thyagarajan.

**Funding acquisition:** Kenneth K. Lau, Dominic Thyagarajan.

**Investigation:** Andrew Ma, Dominic Thyagarajan.

**Methodology:** Andrew Ma, Dominic Thyagarajan.

**Project administration:** Andrew Ma, Dominic Thyagarajan.

**Resources:** Andrew Ma, Kenneth K. Lau, Dominic Thyagarajan.

**Software:** Andrew Ma, Dominic Thyagarajan.

**Supervision:** Kenneth K. Lau, Dominic Thyagarajan.

**Visualization:** Andrew Ma.

**Writing – original draft:** Andrew Ma.

**Writing – review & editing:** Andrew Ma, Kenneth K. Lau, Dominic Thyagarajan.

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
