## [Decision Letter · Decision Letter 0]

7 Jun 2021

PONE-D-21-11514

Radiological correlates of vocal fold bowing as markers of Parkinson’s disease progression: A cross-sectional study utilizing dynamic laryngeal CT

PLOS ONE

Dear Dr. Thyagarajan,

Thank you for submitting your manuscript to PLOS ONE. After careful consideration, we feel that it has merit but does not fully meet PLOS ONE’s publication criteria as it currently stands. Therefore, we invite you to submit a revised version of the manuscript that addresses the points raised during the review process.

Two reviews were submitted, being rather different in the evaluation of the mansuctript.  

Please, especially answer to the major issues raised by reviewer1 who raised concerns on the fundamental experimental procedure to study tremor/Parkinson. Hence, please motivate and justify the performed experimental procedure.

We look forward to receiving your revised manuscript.

Kind regards,

Michael Döllinger, Ph.D.

Academic Editor

PLOS ONE

Additional Editor Comments:

Two reviews were submitted, being rather different in the evaluation of the mansuctript.

Please, especially answer to the major issues raised by reviewer1 who raised concerns on the fundamental experimental procedure to study tremor/Parkinson. Hence, please motivate and justify the performed experimental procedure.

Journal Requirements:

"A. Ma received salary from Ipsen as support for a Movement Disorders Fellowship and received support from an Australian Government Research Training Program (RTP) Scholarship. The funders had no role in study design, data collection and analysis, decision to publish, or preparation of the manuscript." 

We note that one or more of the authors are employed by a commercial company: Ipsen Biopharmaceuticals.

2.1. Please provide an amended Funding Statement declaring this commercial affiliation, as well as a statement regarding the Role of Funders in your study. If the funding organization did not play a role in the study design, data collection and analysis, decision to publish, or preparation of the manuscript and only provided financial support in the form of authors' salaries and/or research materials, please review your statements relating to the author contributions, and ensure you have specifically and accurately indicated the role(s) that these authors had in your study. You can update author roles in the Author Contributions section of the online submission form.

2.2. Please also provide an updated Competing Interests Statement declaring this commercial affiliation along with any other relevant declarations relating to employment, consultancy, patents, products in development, or marketed products, etc.  

Reviewers' comments:

Reviewer's Responses to Questions

**Comments to the Author**

1. Is the manuscript technically sound, and do the data support the conclusions?

Reviewer #1: Yes

Reviewer #2: Yes

2. Has the statistical analysis been performed appropriately and rigorously? 

Reviewer #1: I Don't Know

Reviewer #2: No

3. Have the authors made all data underlying the findings in their manuscript fully available?

Reviewer #1: Yes

Reviewer #2: No

4. Is the manuscript presented in an intelligible fashion and written in standard English?

Reviewer #1: Yes

Reviewer #2: Yes

5. Review Comments to the Author

Reviewer #1: In the material and methods: Line 89-93

“During a continuous CT acquisition scanning period of 5 seconds over an anatomical z-axis of 16 cm over the larynx without CT table movement, participants were instructed to produce five short phonations of /i/ quickly and clearly.”

It meant that the subjects produced many times short phonation of /i/?

In general, both aged healthy subjects and pwPD subjects showed adductor muscles atrophy, which make clinically vocal fold atrophy and glottal incompetence.

The difference of phonatory function is with or without a voice tremor. In pwPD, the most important symptom is the voice tremor.

Authors should better use sustain phonation as a phonatory task. Sustain phonation /a/, may identify tremor on pwPD patients in dynamic CT study.

In the summery or conclusion in abstracts: line 39-42

“The increases in GA despite reductions in IAD are concordant with prior observations of vocal fold bowing. Our study provides a basis for using laryngeal 4D-CT to assess disease progression in Parkinson’s disease.”

Vocal fold bowing can observe not only in pwPD, but also aged healthy subjects because it is just physiological change of aging caused by vocal fold atrophy. However, GA and IAD are definitely correlate with the glottal chink because of glottal incompetence.

In clinical situation, observation with laryngeal fiberscope and maximum phonation time (MPT), and acoustical analysis is more useful to assess disease progression in Parkinson’s disease. Furthermore, those are less invasive. Authors might better study those relationship.

I am afraid to say, authors should better mention about voice tremor in PD patients, and the difference of the maximum phonation time in each patient and subject.

Reviewer #2: Dear authors,

Thank you for preparing this interesting and well-written manuscript. The study certainly adds new information to the understanding of the larynx anatomy and the vocal fold movements during phonation in patients with Parkinson’s Disease (PD). The aim was to test a new way to detect changes in the larynx related to the duration and severity of the disease. You often refer to a previous study, but some more information needs to be given in the present manuscript for the reader to be able to understand your work.

Abstract OK

Introduction:

I clear and focused leading to the aims.

Page 3, line 57, I suggest that you add “yet” after “…dysfunction) have not YET found a place… “. There may be acoustic measures available in the future for measuring dysphonia severity in patients with PD.

Material and methods

Page 4. Line 73. Please add information about how the patients were recruited and if they had had any previous voice training, such as LSVT. That information is relevant since behavioral treatment can possibly affect the glottal closure during phonation.

Page 4, line 75-77. There Is an overlap in duration between the two pw groups, “less than 6 years” and “5 year or greater”. Please comment on that and why you choose the criteria (besides the H&Y data).

Page 4, A paragraph about the procedure is missing. Please add information about the time it took for the recordings, the instructions given to the patients, did they practice?

Page 5, line 91-92. Please add the rationale for choosing 5 repetitions of short /i/ sounds. What was quick and clearly mean related to loudness and pitch? Describe how the phonation varied between the 5 repetitions?

Did you have any problems with data analysis leading to missing values?

Statistical analysis:

Page 5, line 100-101. Why did you remove all data from the pre-vocalization? You refer to your previous study, however, you cannot expect each reader to read your previous study.

Page 5, line 103. So how many timepoints in total did you base your findings on?

Page 5, lines 110-114. Did you measure the movements separate for the left and right vocal folds? In the interesting discussion you bring up the possible side-differences.

Page 6, is the first paragraph the Figure Caption for Figure 1?

Page 6, line 129-131. Please describe the variation in the material (descriptive statistics) and not only the Md.

Page 6, line 138. There is no information in the introduction about age and sex. Did you expect differences in the results related to age and gender? If so, please add information in the Introduction. It may be of interest since the dimensions and position of the laryngeal cartilages related to the length of the vocal folds may affect glottal closure during phonation differently in male and female speakers.

Results

The result section (from page 7 to 11) is mainly written as figure captions as I interpret the text. Please make the text body different from the Figure captions.

Page 7. It will be great when the raw data will be available.

You performed many statistical tests, please motivate your choice of p-value.

Page 8, line 165-174. Is this paragraph a caption to Figure 2?

Figure 3. Why do you need to write all the p-values? The scattergrams speak for themselves. Even for the highest correlation coefficient (r=.692) you can see quite a spread in the data.

Figure 3. The units are missing for Mean IAI.

Discussion

Page 11, line 210. Good that you express yourself carefully that “ these laryngeal measures have a potential utility as a marker… ”. Especially since the correlations are not specifically high in Figure 3.

You interpret your findings in a meaningful and pedagogical way in Figure 4 that is clear.

Page 12, line 234. You refer to “The authors of that study….” It is not written to what study you refer.

Page 12, line 235. It is very interesting with the eventual side difference in muscle contraction. So did you measure any side differences in your material. If that is true, do you have any thoughts how this would affect phonation and voice quality?

Page 14, line 287-292. Regarding limitations, there is a need that you also discuss the variation on your material related to phonation. It would be interesting to know details about how the patients performed the phonation task and if that affected the measures, such as loud vs soft phonation that is known to affect glottal adduction and closure.

Page 14, line 297-298. Validating the results in a prospective study with other imaging modalities, that you suggest, is of course needed. And if ultrasound would be available in the future that of course would be great.

There is lack of information about the relationship between your findings and voice acoustics. Please discuss if you think it would be of value to include perceptual and acoustic measures as well in a validation study? To correlate the voice characteristics to underlying anatomy may help to increase the reliability and validity in the acoustic and perceptual assessment that is problematic, but still important in the clinic. The laryngeal measures, that you have identified to correlate with progression of PD are highly relevant because of its role in phonation too.

Conclusions are missing in the end of the Discussion

6. PLOS authors have the option to publish the peer review history of their article (what does this mean?). If published, this will include your full peer review and any attached files.

Reviewer #1: No

Reviewer #2: No

---

## [Author Response · Author response to Decision Letter 0]

24 Jun 2021

To the editor and reviewers,

Thank you all greatly for your time, consideration, and insightful comments on our research article, ‘Radiological correlates of vocal fold bowing as markers of Parkinson’s disease progression: A cross-sectional study utilizing dynamic laryngeal CT’. We are now re-submitting the manuscript accordingly in line with many of the critiques and suggestions put forward, which we have edited in with tracked changes.

We have acknowledged your comments (quoted in grey) and would like to respond to these in kind to indicate the changes we have since made (quoted in blue), or to clarify our existing methods or statements.

Journal Requirements

We have updated formatting, particularly with respect to headings and references, to conform with the style requirements.

2.1. Please provide an amended Funding Statement declaring this commercial affiliation, as well as a statement regarding the Role of Funders in your study.

We would like to update our Funding Statement as follows:

“A. Ma received salary from Ipsen Biopharmaceuticals as support for a Movement Disorders Fellowship and received support from an Australian Government Research Training Program (RTP) Scholarship. The funders provided support in the form of salaries for author AM, but did not have any additional role in the study design, data collection and analysis, decision to publish, or preparation of the manuscript. The specific roles of these authors are articulated in the ‘author contributions’ section.”

2.2. Please also provide an updated Competing Interests Statement declaring this commercial affiliation along with any other relevant declarations relating to employment, consultancy, patents, products in development, or marketed products, etc.  

We would like to update our Competing Interests Statement as follows:

“A. Ma received salary from Ipsen Biopharmaceuticals as support for a Movement Disorders Fellowship and received support from an Australian Government Research Training Program (RTP) Scholarship. The funders had no role in study design, data collection and analysis, decision to publish, or preparation of the manuscript. D. Thyagarajan and K.K. Lau report no disclosures relevant to the manuscript. This does not alter our adherence to PLOS ONE policies on sharing data and materials.”

Reviewer #1

It meant that the subjects produced many times short phonation of /i/?

We have clarified in the manuscript that subjects produced five short phonations of /i/, which is now as follows:

Line number 97-99: “During a continuous CT acquisition scanning period of 5 seconds, participants were instructed to produce five short phonations of /i/ quickly and clearly at a comfortable speaking volume and pitch.”

Authors should better use sustain phonation as a phonatory task. Sustain phonation /a/, may identify tremor on pwPD patients in dynamic CT study...The difference of phonatory function is with or without a voice tremor. In pwPD, the most important symptom is the voice tremor… I am afraid to say, authors should better mention about voice tremor in PD patients.

We agree that vocal tremor would be a potentially interesting line of enquiry and could be looked into in a future study. Indeed some patients with Parkinson’s disease have vocal tremor, with a prevalence ranging between 13-68% on perceptual studies and 14.6-55% on endoscopic studies. However, on acoustic analysis, vocal tremor was detected at a similar prevalence in pwPD (43%) to healthy controls (36%). 

When studying vocal tremor, we agree that a sustained phonatory task is the most appropriate. However the focus of this study was arytenoid cartilage movements, not vocal tremor, and hence the choice of a repeated phonation task. The reviewer’s comments raise the possibility that vocal tremor may account for some of the variance in our results. We thank the reviewer and have modified the manuscript at line ### to address the comments of the reviewer:

Line number 309-315: “The presence of vocal tremor was also not assessed. Vocal tremor occurs in some pwPD, with a frequency depending on technique and definition – 13-68% on perceptual studies and about 15-55% on endoscopic studies – but at a similar prevalence to controls on acoustic analysis [21]. The contribution of vocal tremor is therefore uncertain. The addition of a sustained phonation task may have helped us to understand the contribution of vocal tremor, but this would have exposed the subjects to more radiation.”

In general, both aged healthy subjects and pwPD subjects showed adductor muscles atrophy, which make clinically vocal fold atrophy and glottal incompetence…Vocal fold bowing can observe not only in pwPD, but also aged healthy subjects because it is just physiological change of aging caused by vocal fold atrophy. However, GA and IAD are definitely correlate with the glottal chink because of glottal incompetence.

We thank the reviewer for this observation. We have modified the manuscript accordingly to highlight the fact that bowing occurs in healthy ageing as well. We expect that the inclusion of age-matched controls would account for this phenomenon.

Line number 278-280: “Although vocal fold bowing can also occur with ageing due to vocal fold atrophy [20], we showed that the mean/maximum IAI is increased in pwPD when compared to age-matched controls.”

In clinical situation, observation with laryngeal fiberscope and maximum phonation time (MPT), and acoustical analysis is more useful to assess disease progression in Parkinson’s disease. Furthermore, those are less invasive. Authors might better study those relationship… 

We are not aware of high-quality prospective longitudinal studies which correlate acoustic analysis or laryngoscopy with disease progression. Certainly, observational studies have been performed in Parkinson’s disease. However not all of these are adequately controlled or are quantitative by design. We agree with the reviewer that maximum phonation time, acoustic analysis and laryngoscopy are useful clinical investigations, but we were interested in studying the arytenoid cartilage movements in this disorder of movement. Hence, our interest in investigating by a quantitative and controlled study using the novel modality of dynamic four-dimensional laryngeal CT, particularly in relation to how Parkinson’s disease changes over time and severity. 

I am afraid to say, authors should better mention…the difference of the maximum phonation time in each patient and subject.

We thank the reviewer for raising the importance of maximum phonation time (MPT). We have studied MPT and other perceptual variables in our previous paper on four-dimensional CT (Perju-Dumbrava et al, 2017). Indeed, we found that MPT was significantly shorter for patients with Parkinson’s disease. However, in this paper, we were concerned mainly with how arytenoid movements changed with disease progression. Perceptual measures including MPT were not a focus of this paper but we agree with the reviewer that it would also be interesting to study how these parameters would change with disease progression in an appropriately controlled trial.

We have modified the manuscript in our Introduction to mention the MPT as the reviewer has recommended:

Line number 62-66: “In that study, participants also underwent perceptual analysis and we found significant increases in breathiness and articulatory dysdiadochokinesis, and reductions in loudness variability and mean phonation time in Parkinson’s disease. We found no effect of age or sex on IAD or GA.” 

Reviewer #2

Introduction

Page 3, line 57, I suggest that you add “yet” after “…dysfunction) have not YET found a place… “. There may be acoustic measures available in the future for measuring dysphonia severity in patients with PD.

Thank you for the suggestion. We have modified the manuscript accordingly as follows:

Line number 56-58: “Despite this, perceptual, acoustic or other measures of hypokinetic dysarthrophonia in PD (e.g. vocal cord dysfunction) have not yet found a place in the assessment of disease progression.”

Material and methods

Page 4. Line 73. Please add information about how the patients were recruited…

We have added in information to the methods section to clarify the recruitment process as follows:

Line number 79-86: “We expanded this cohort by recruiting patients with more advanced PD from the Movement Disorders Clinic at Monash Medical Centre… Control participants either responded to advertisements or were spouses of the pwPD who were unaffected by Parkinson’s disease or other neurological disorders apparent on clinical assessment by a neurologist.”

Please add information about…if they had had any previous voice training, such as LSVT. That information is relevant since behavioral treatment can possibly affect the glottal closure during phonation.

We appreciate this insightful comment on an important factor which we had not mentioned. We have now included the following:

Line number 163: “No pwPD recruited had previously undergone voice training.”

Page 4, line 75-77. There Is an overlap in duration between the two pw groups, “less than 6 years” and “5 year or greater”. Please comment on that and why you choose the criteria (besides the H&Y data).

Unfortunately, there is no consensus in the literature on what constitutes early or more advanced disease. We therefore intended to avoid dichotomising the disease as ‘early’ and ‘late’ in our analysis. Instead, we have treated our entire study population as a single cohort with disease duration considered as a continuous variable. The choice of recruitment criteria overlapping in duration with that of the first study was to recruit adequately across the spectrum of disease duration and severity. We thank the reviewer for picking this up and we have struck out the term ‘early disease’ from our materials and methods and clarified the rationale for having overlapping criteria for disease duration as follows:

Line number 80-83: “For the purposes of obtaining adequate representation across the spectrum of the disease, we recruited patients who were H&Y stage 3 or more and a disease duration of 5 years or greater.”

Page 4, A paragraph about the procedure is missing. Please add information about the time it took for the recordings, the instructions given to the patients, did they practice?...What was quick and clearly mean related to loudness and pitch?

Thank you for raising this. Patients were imaged over 5 seconds. Instructions were provided to patients to vocalize quickly and clearly at a comfortable speaking pitch and volume. They were trained in the task prior to image acquisition. We have added clarifications to our methods section accordingly as follows: 

Line number 98-101: “…participants were instructed to produce five short phonations of /i/ quickly and clearly at a comfortable speaking volume and pitch. Patients practiced the vocalization task prior to undergoing the scan. Multiple phonations were performed to allow study of the arytenoid cartilages in motion in as repeatable a way as possible.”

Page 5, line 91-92. Please add the rationale for choosing 5 repetitions of short /i/ sounds…Describe how the phonation varied between the 5 repetitions?

We chose a repeated phonation task in order to study arytenoid cartilage movements during vocalization in as repeatable (within and between subjects) a way as possible. We could not measure how phonation varied as we could not record participants during image acquisition. We thank you for bringing this up and have acknowledged this in our discussion as a limitation of the study:

Line number 304-308: “Limitations of our study include the possibility of variance being introduced by differences amongst participants in volume, pitch and cadence. Vocalization during image acquisition could not be recorded due to the loudness of the scanning apparatus. As such, we could not account for these acoustic parameters which may influence arytenoid cartilage dynamics.” 

Did you have any problems with data analysis leading to missing values?

We excluded from analysis the patients described in line number ###. Missing values due to data trimming did not affect the analysis as these are handled by the linear mixed model analysis which accounts for differences in the vocalization time. 

Statistical analysis

Page 5, line 100-101. Why did you remove all data from the pre-vocalization? You refer to your previous study, however, you cannot expect each reader to read your previous study.

The non-vocalization period contaminates what is occurring during the period of study (vocalization). We have now included the methodology of data trimming and rationale for this:

Line number 111-116: “In pre-processing, we manually removed IAD and GA values from the pre and post-vocalization period. Whilst not vocalizing, values of both the GA and IAD are strikingly higher as the vocal folds are apart. Trimming was performed by visually inspecting the raw GA and IAD data sets and excluding periods containing these larger values. This was done as the study was focused on assessing the movement and posturing of the vocal folds during vocalization.”

Page 5, line 103. So how many timepoints in total did you base your findings on?

This information is now provided as follows:

Line number 167-168: “After exclusion of these patients and data trimming, the total number of timepoints analysed was 1671 for the IAD, 1705 for the GA and 1607 for the IAI.”

Page 5, lines 110-114. Did you measure the movements separate for the left and right vocal folds? In the interesting discussion you bring up the possible side-differences.

We mentioned the asymmetry of arytenoid cartilage posturing in the discussion to describe the previous observations on laryngoscopy which showed hyperadduction of the vocal folds. We did out of interest an analysis on the side-to-side differences in the arytenoid cartilage movement between timepoints and did not find any asymmetry. However, as the study was not primarily designed to assess symmetry of arytenoid cartilage movement, and such baseline characteristics as to asymmetry of body involvement were not collected, we did not include this analysis in our manuscript.

Page 6, line 129-131. Please describe the variation in the material (descriptive statistics) and not only the Md.

Descriptive statistics for the minimum IAD and GA, maximum IAI and standard deviation for each of these laryngeal measures across the vocalization period are now provided in supplementary table 1. In creating this table, it became apparent that referring to these derived minima and maxima in this manner could be confusing, and we now refer to these as the ‘effective minimum’ and ‘effective maximum’ throughout the manuscript and figures. We have uploaded updated figures, and also made the following changes to the manuscript to clarify this terminology and rationale for doing so:

Line number 140-143: " We defined an ‘effective minimum’ for IAD and GA as the median of the lowest five values, as the vocalization task involved phonating /i/ five times. Similarly, the ‘effective maximum’ IAI, AS, and was defined as the median of the highest five values. This was done to minimise the potential distorting effects of outliers.”

Page 6, line 138. There is no information in the introduction about age and sex. Did you expect differences in the results related to age and gender? If so, please add information in the Introduction. It may be of interest since the dimensions and position of the laryngeal cartilages related to the length of the vocal folds may affect glottal closure during phonation differently in male and female speakers.

Whilst there are perceptual and acoustic differences in voice between men and women, and with varying age, we did not expect differences in arytenoid cartilage movement based on the results of our previous study. In this study also, we tested the effect of age and sex on IAD and GA as fixed effects in our model and neither was statistically significant (see supplementary table 2). We thank you for bringing this up and have now included this information into the Introduction as follows:

Line number 65-66: “We found no effect of age or sex on IAD or GA.”

Results

The result section (from page 7 to 11) is mainly written as figure captions as I interpret the

text. Please make the text body different from the Figure captions…Page 6, is the first paragraph the Figure Caption for Figure 1?...Page 8, line 165-174. Is this paragraph a caption to Figure 2?

The paragraphs following the figure captions were legends for the respective figures as the reviewer had determined. They have now been placed in line with the figure title in the manuscript, which is in keeping with the formatting instructions for PLOS One (https://journals.plos.org/plosone/s/file?id=9cba/PLOS%20Manuscript%20Body%20Formatting%20Guidelines.pdf).

Page 7. It will be great when the raw data will be available.

The raw data is available for viewing currently at https://figshare.com/s/e69d4ea6a8f4bc031b7f. The public link currently in the manuscript will be activated upon acceptance. 

You performed many statistical tests, please motivate your choice of p-value.

In line with recent recommendations, we have included exact p-values where appropriate. We have interpreted ‘motivate your choice of p-value’ to indicate that we have not explicitly stated our alpha, and we have now indicated an alpha level of 0.05 as is convention as follows:

Line number 146-147: “We set an alpha level of 0.05 for all statistical tests.”

Figure 3. Why do you need to write all the p-values? The scattergrams speak for themselves. Even for the highest correlation coefficient (r=.692) you can see quite a spread in the data.

Thank you for your comments here. We agree with the comment and we have edited the figure to indicate the significance level of each linear model with asterisks. 

Figure 3. The units are missing for Mean IAI.

Due to the way IAI is calculated (square of a distance divided by area), it is a dimensionless index and does not have a corresponding unit.

Discussion

Page 11, line 210. Good that you express yourself carefully that “ these laryngeal measures have a potential utility as a marker… ”. Especially since the correlations are not specifically high in Figure 3… You interpret your findings in a meaningful and pedagogical way in Figure 4 that is clear.

Thank you.

Page 12, line 234. You refer to “The authors of that study….” It is not written to what study you refer.

Thank you. We have changed the sentence structure for clarity. It now reads:

Line number 249-253: “This is in keeping with laryngoscopic observations that in pwPD whose motor symptoms are asymmetric, on the side of the body with more severe parkinsonism, the arytenoid cartilage is hyperadducted, its vocal fold closes underneath the contralateral vocal fold and there is increased contraction of the adductor muscles of the vocal folds.”

Page 12, line 235. It is very interesting with the eventual side difference in muscle contraction. So did you measure any side differences in your material. If that is true, do you have any thoughts how this would affect phonation and voice quality?

As mentioned above, we analysed the side-to-side difference in arytenoid cartilage movements out of interest, although this was not an initial intention of the study. While we did not detect a difference in our data, we hypothesise that the asymmetry of vocal fold posturing and movement observed on laryngoscopy leads to asymmetry in the tension of the vocal folds. This would lead to a disparity in the vibratory dynamics of each fold. We wonder if this could give rise to some aspects of the dysphonia of Parkinson’s disease. 

Page 14, line 287-292. Regarding limitations, there is a need that you also discuss the variation on your material related to phonation. It would be interesting to know details about how the patients performed the phonation task and if that affected the measures, such as loud vs soft phonation that is known to affect glottal adduction and closure.

Thank you. We agree that this information would be helpful, and we had tried to obtain recordings of the vocalizations made during image acquisition. Unfortunately, the loudness of the scanning apparatus did not permit this. We have added this information into the discussion of our limitations as follows:

Line number 304-308: “Limitations of our study include the possibility of variance being introduced by differences amongst participants in volume, pitch and cadence. Vocalization during image acquisition could not be recorded due to the loudness of the scanning apparatus. As such, we could not account for these acoustic parameters which may influence arytenoid cartilage dynamics.” 

Page 14, line 297-298. Validating the results in a prospective study with other imaging modalities, that you suggest, is of course needed. And if ultrasound would be available in the future that of course would be great.

Thank you for your kind comments. A prospective study with 4D laryngeal CT is in part limited by the need for repeated radiation exposure. We hope that ultrasound may provide a means of studying these movements in a prospective manner. 

There is lack of information about the relationship between your findings and voice acoustics. Please discuss if you think it would be of value to include perceptual and acoustic measures as well in a validation study? To correlate the voice characteristics to underlying anatomy may help to increase the reliability and validity in the acoustic and perceptual assessment that is problematic, but still important in the clinic. The laryngeal measures, that you have identified to correlate with progression of PD are highly relevant because of its role in phonation too.

We studied the correlation between our laryngeal measures and perceptual measures of voice in our previous study, and only weak or non-significant correlations were seen. These perceptual tests were hence not assessed again on the participants recruited for this paper. We do agree that acoustic analysis is of utmost importance; however it fell outside the scope of this study which focused on assessing the changes to arytenoid cartilage movements with the progression of Parkinson’s disease. 

Conclusions are missing in the end of the Discussion

We have not included a formal conclusion section as it is optional according to the instructions for authors: “The following elements can be renamed as needed and presented in any order… Conclusions (optional)” at https://journals.plos.org/plosone/s/submission-guidelines.

However, we have addressed the reviewer’s comment by altering the discussion section thus:

Line number 322-325: “In summary, this study has identified radiographic measures which correlate with the anatomic changes previously observed in laryngoscopic studies of pwPD. We conclude that certain measures of laryngeal dynamics change with increasing duration and severity of PD.” 

Thank you all once again for your time and consideration of our manuscript. 

Yours sincerely,

---

## [Decision Letter · Decision Letter 1]

26 Jul 2021

PONE-D-21-11514R1

Radiological correlates of vocal fold bowing as markers of Parkinson’s disease progression: A cross-sectional study utilizing dynamic laryngeal CT

PLOS ONE

Dear Dr. Thyagarajan,

Thank you for submitting your manuscript to PLOS ONE. After careful consideration, we feel that it has merit but does not fully meet PLOS ONE’s publication criteria as it currently stands. Therefore, we invite you to submit a revised version of the manuscript that addresses the points raised during the review process.

The authors apply Linear mixed model statistics (line148): please specify which specific model was used and report that the necessary requirements were met – this is unfortunately not given nor provided in the text body or Table S2. Also, which software was used to perform statistics?

We look forward to receiving your revised manuscript.

Kind regards,

Michael Döllinger, Ph.D.

Academic Editor

PLOS ONE

Journal Requirements:

Additional Editor Comments (if provided):

The authors apply Linear mixed model statistics (line148, 213 and following): please specify which specific model was used and report that the necessary requirements were met – this is unfortunately not given nor provided in the text body or Table S2. Also, which software was used to perform statistics?

Reviewers' comments:

Reviewer's Responses to Questions

**Comments to the Author**

1. If the authors have adequately addressed your comments raised in a previous round of review and you feel that this manuscript is now acceptable for publication, you may indicate that here to bypass the “Comments to the Author” section, enter your conflict of interest statement in the “Confidential to Editor” section, and submit your "Accept" recommendation.

Reviewer #2: All comments have been addressed

Reviewer #3: All comments have been addressed

2. Is the manuscript technically sound, and do the data support the conclusions?

Reviewer #2: Yes

Reviewer #3: Yes

3. Has the statistical analysis been performed appropriately and rigorously? 

Reviewer #2: Yes

Reviewer #3: Yes

4. Have the authors made all data underlying the findings in their manuscript fully available?

Reviewer #2: Yes

Reviewer #3: Yes

5. Is the manuscript presented in an intelligible fashion and written in standard English?

Reviewer #2: Yes

Reviewer #3: Yes

6. Review Comments to the Author

Reviewer #2: Dear Authors,

I have now read the revised version of your manuscript and have found that you have taken all comments into consideration and changed the text accordingly.

I just want you to know that I have a clinical reflection since I was quite astonished that none of the pwPD in your study cohort had received or been offered voice therapy. This was not within the scope of your study, of course, but I found this strange, since many patients can improve their voice function with e g., LSVT.

Kind regards

Reviewer #3: The premise of the study is that the laryngeal measures used change as PD progresses. The patient cohort expands on the previous referenced project, and includes PD patients with longer disease duration. It uses a dynamic four-dimensional laryngeal CT and quantitative anatomical assessments to correlate the radiographic measures with the reported observed changes during laryngoscopy.

The authors appear to have satisfactorily addressed the comments of the 2 reviewers. Limitations of their study are acknowledged and they have been measured in their comments about the usefulness of this method in clinical practice.

This study is focussed on a particular vocal motor task with a set number of measures, and has to be interpreted appropriately. The analysis is described in sufficient detail and to an appropriate technical standard. The conclusions are clear and supported by the data. The study meets suitability for publication.

As the authors mention, this study is a next step in the search of objective measures of movements in PD patients in an area that can be affected early in the disease course and will hopefully lead to further research.

7. PLOS authors have the option to publish the peer review history of their article (what does this mean?). If published, this will include your full peer review and any attached files.

Reviewer #2: No

Reviewer #3: No

---

## [Author Response · Author response to Decision Letter 1]

6 Sep 2021

Comment to Editor:

"Additional Editor Comments (if provided):

The authors apply Linear mixed model statistics (line148, 213 and following): please specify which specific model was used and report that the necessary requirements were met – this is unfortunately not given nor provided in the text body or Table S2. Also, which software was used to perform statistics? "

Response:

Thank you for raising this issue. We have now clarified our preferred model in the results section as follows:

Line number 222-227: “The transformed values of the IAI in model 3, predicting the duration (b = 0.0054, SEb = 0.0014, 95% CI [0.0024, 0.0083], t(27) = 3.69, p < .001) and model 6, predicting severity (b = 0.031, SEb = 0.0089, 95% CI [0.012, 0.049], t(27) = 3.41, p = 0.002) were the preferred models for interpretation based on having the highest marginal R-squared values (see S2 Table for model details and comparisons).”

We considered age and sex to be clinically relevant fixed factors in the model and chose to retain these in the models we reported in S2 Table. When individual factors were dropped, the AIC and BIC values did not change very much so we chose to report the most clinically relevant models described in S2 Table. 

Edits were also made to the better align with APA guidelines for reporting the statistical parameters of the IAD models in line number 219-221 and in the abstract in line number 36-37.

We also clarified the absence of significant interactions in the model terms as follows:

Line number 229-230: “Additionally, no significant interaction was found between duration of disease or severity and participant age or sex.”

We also altered the discussion section to reflect the preference for the IAI:

Line number 289-291: “The preferred linear mixed models based on effect size were those which treated transformed values of the IAI to explain disease duration or severity.”

…and report that the necessary requirements were met – this is unfortunately not given nor provided in the text body or Table S2.

We had tested for the standard statistical assumptions. We thank you for bringing to our attention that this was not made clear in the manuscript. We have now included the following to clarify this:

Line number 216-218: “Non-linear transformations were applied to the GA and IAI to achieve linearity of the data set. This allowed all models to meet all the standard statistical assumptions.”

---

## [Editor Report · Decision Letter 2]

6 Oct 2021

Radiological correlates of vocal fold bowing as markers of Parkinson’s disease progression: A cross-sectional study utilizing dynamic laryngeal CT

PONE-D-21-11514R2

Dear Dr. Thyagarajan,

We’re pleased to inform you that your manuscript has been judged scientifically suitable for publication and will be formally accepted for publication once it meets all outstanding technical requirements.

Kind regards,

Michael Döllinger, Ph.D.

Academic Editor

PLOS ONE
---

## [Editor Report · Acceptance letter]

8 Oct 2021

PONE-D-21-11514R2 

Radiological correlates of vocal fold bowing as markers of Parkinson’s disease progression: A cross-sectional study utilizing dynamic laryngeal CT 

Dear Dr. Thyagarajan:

I'm pleased to inform you that your manuscript has been deemed suitable for publication in PLOS ONE. Congratulations! Your manuscript is now with our production department. 

Kind regards, 

on behalf of

Dr. Michael Döllinger 

Academic Editor

PLOS ONE